# Development of a Hot-Melt-Extrusion-Based Spinning Process to Produce Pharmaceutical Fibers and Yarns

**DOI:** 10.3390/pharmaceutics14061229

**Published:** 2022-06-10

**Authors:** Christoph Rosenbaum, Linus Großmann, Ellen Neumann, Petra Jungfleisch, Emre Türeli, Werner Weitschies

**Affiliations:** 1Department of Biopharmaceutics and Pharmaceutical Technology, Institute of Pharmacy, University of Greifswald, Felix-Hausdorff-Straße 3, 17489 Greifswald, Germany; christoph.rosenbaum@uni-greifswald.de (C.R.); linus.grossmann@uni-greifswald.de (L.G.); ellen.neumann@stud.uni-greifswald.de (E.N.); 2Mybiotech GmbH, Industriestraße 1b, 66802 Überherrn, Germany; p.jungfleisch@mybiotech.de (P.J.); e.tuereli@mybiotech.de (E.T.)

**Keywords:** Germany, formulation design, manufacturing science, materials science, hotmelt extrusion, spinning process, continuous manufacturing

## Abstract

Fibers and yarns are part of everyday life. So far, fibers that are also used pharmaceutically have mainly been produced by electrospinning. The common use of spinning oils and the excipients they contain, in connection with production by melt extrusion, poses a regulatory challenge for pharmaceutically usable fibers. In this publication, a newly developed small-scale direct-spinning melt extrusion system is described, and the pharmaceutically useful polyvinyl filaments produced with it are characterized. The major parts of the system were newly developed or extensively modified and manufactured cost-effectively within a short time using rapid prototyping (3D printing) from various materials. For example, a stainless-steel spinneret was developed in a splice design for a table-top melt extrusion system that can be used in the pharmaceutical industry. The direct processing of the extruded fibers was made possible by a spinning system developed called Spinning-Rosi, which operates continuously and directly in the extrusion process and eliminates the need for spinning oils. In order to prevent instabilities in the product, further modifications were also made to the process, such as a the moisture encapsulation of the melt extrusion line at certain points, which resulted in a bubble-free extrudate with high tensile strength, even in a melt extrusion line without built-in venting.

## 1. Introduction

Fiber and thread materials are part of everyday life, including everyday clothing, the highly engineered fibers in airbags and seat belts in cars, and sterile and life-saving suture material for surgery [1,2,3]. The pharmaceutical application of fibers, especially materials produced by electrospinning, have also been described [4,5,6,7]. In the past, various systems for the production of materials similar to fibers and yarns were developed and their descriptions were published [8]. Essentially, a distinction can be made between melt spinning and solution spinning. In melt spinning, a powder or granular material is melted and then spun [9]. Solution spinning is subdivided into dry spinning, wet spinning, gel spinning, and electrospinning [6,7,10]. These processes are used, in particular, for polymers that already decompose at low temperatures. In electrospinning, a polymer solution is usually metered to an electrode in an electric field. The solution, accelerated by the electric field toward another electrode, is stretched down into very fine fibers, which are deposited on the counter-electrode to form a kind of fleece [6,11]. The process has already been used in various works for the production of pharmaceutical dosage forms, but is not especially productive. It is therefore only suitable for particular specialty products.

In the melt spinning process, which is particularly suitable for synthetic fibers, the starting materials are liquefied by means of heat and then pressed through a special spinneret [9]. The number of holes in a spinneret is decisive for the number of individual filaments in a yarn braid and, thus, for the properties of the resulting yarn. Monofilaments are created by means of only one hole in the spinneret, whereas multifilaments in the technical sector have up to 250,000 individual filaments [9,12]. The melt emerging from the spinneret is then further processed by pulling it into extremely fine individual fibers. In addition, a so-called avivage is applied to the fibers in the same manufacturing step [13]. The avivage often consists of fatty-acid polyglycol esters, glycol ether mixtures, fatty-acid condensate products, dialkyl polyglycol ethers, oxyalkylated fatty-acid derivatives, phosphoric acid esters, mineral oils, wax dispersions, or quaternary amines, and leads to increased smoothness or enables further processing due to the necessary silkiness and the prevention of possible electrostatic behavior [13]. However, as advantageous as the lubrication of fiber materials is in the production of technical fibers, it can be problematic, particularly due to the excipients used, which are not approved or described for pharmaceutical applications [14,15]. Following the production of technical fibers, they are further treated and, usually, a washing process is also implemented in the final manufacturing process [13]. In this process, suitable cleaning agents and solvents are used to wash off components of the spinning oils. The use of solvents can also be a challenge for pharmaceutical applications and may further complicate approval [16,17].

Until now, the use of fiber materials has been limited in the pharmaceutical industry, but they are ubiquitous in medical technology [3]. Special fibers are used, for example, as self-degrading surgical suture materials or to perform esophageal biopsies with the ingestion of cytosponges [7,18]. In the area of drug therapy, for example, it is necessary to mention the recently developed EsoCap system, in which a thread is needed to place a film in the esophagus [19]. Further applications in the sector of drug therapy using, for example, hot-melt-extruded fibers loaded with active ingredients are conceivable, and might create new possibilities and methods for the application of active ingredients [20,21]. Furthermore, in the field of dressing materials, the functionality of a large number of fiber- or textile-based products could be extended by using, for example, disinfecting or anti-inflammatory active ingredients [22,23,24].

The development of innovative dosage forms requires new concepts. In this regard, the field of fiber-based dosage forms could be of significant interest. To avoid the aforementioned challenges in the production of fiber and yarn materials as far as possible, a manufacturing process needs to be developed that allows the production of pharmaceutical fibers in principle. For this purpose, existing techniques already established in the field of pharmaceutical melt extrusion need to be used, if possible, and optimized, modified or, if necessary, extended in accordance with the objective. All processes should be designed as cost-effectively as possible, ideally using 3D printing, and should have a modular structure to enable potential upscaling and in-line modification. In any case, the use of additional auxiliary materials, such as spinning oils, should be avoided, and plant engineering solutions should be developed. The stability and properties of the fibers and yarns developed should be investigated.

## 2. Materials and Methods

### 2.1. Materials

Polyvinyl alcohol Parteck MXP, as with the plasticizer glycerol anhydrous, was purchased from Merck (Darmstadt, Germany).

### 2.2. Construction and Development of a Spinning Apparatus for Pharmaceutically Used Fibers and Yarns

The system for the production of fibers and yarns can be divided into three different areas: melt extrusion (HME), stretching, and spinning (Figure 1). The melt extrusion line consisted in the ZD 9 FB metering unit from Three-Tec (Sion, Switzerland), which was intended to ensure uniform feeding of material into the extruder. Before use, the polymer used was homogenized by hand with plasticizer in a stainless steel bowl using a pistil, passed through a sieve with a mesh size of 500 µm, and then dried for 3 h at 105 °C in an FD 53 convection-drying oven from Binder (Tuttlingen, Germany). In addition, the metering device and the transition to the extruder were encapsulated due to moisture problems during subsequent extrusion and, moreover, it was possible to add desiccant to the self-constructed encapsulation. The Moxx thermohygrometers from TFA Dostmann (Wertheim-Reicholzheim, Germany), which were installed in the encapsulation, enabled monitoring of the humidity inside the encapsulation. As soon as the air humidity inside the encapsulation was more than 20% r.h., it was possible to replace the desiccant. The components needed to encapsulate the flat-bottomed dispenser and the transition between it and the melt extrusion system were designed individually using the open-source FreeCAD 0.18 software, sliced using the also freely available software Cura 4.4.0 (Ultimaker, Utrecht, The Netherlands), and printed in-house using the 3D printer Ultimaker 3 (Ultimaker, Utrecht, The Netherlands). EasyFil PLA (FormFutura, Nijmegen, The Netherlands), a polymer made from food-grade polylactic acid (PLA), was used for printing.

The Micro-Lite Twin-Screw extruder used (Rondol Industrie SAS, Nancy, France) was equipped with water inlet cooling and segmented screws (Figure 2). Instead of an otherwise standard die plate, we added a special self-constructed multifilament spinneret made of stainless steel and unwound by 90°, manufactured by Protiq (Blomberg, Germany), under contract, using 3D powder bed fusion printing (PBF). Subsequently, the contact surfaces of the spinneret also had to be reworked.

To stretch the extrudate, a 100 × 473 compl. V2 conveyor belt from Three-Tec (Sion, Switzerland), modified to include a stainless-steel pressure roller, was used. All the components required for assembly were self-constructed and created from PLA using 3D printing. The spinning apparatus was also self-constructed, and it was manufactured by means of 3D printing. The spinning wheel was driven by a stepper motor NEMA 24 version PD4-N6018L4204 (Nanotec Electronic, Feldkirchen, Germany), which was controlled by means of the software, NanoPro 1.70.8.0 (Nanotec Electronic, Feldkirchen, Germany). A food-grade polyester yarn (Westmark, Lennenstadt, Germany) was used for pre-feeding or starting the system and as a brake.

### 2.3. Residual Moisture Determination

The residual moisture of the polymer batches to be extruded was investigated using the infrared residual moisture analyzer type MB35 (Ohaus Europe GmbH, Nänikon, Switzerland).

### 2.4. Optical Analysis

Using a Stemi 2000-C microscope (Carl Zeiss, Oberkochen, Germany) in combination with the illumination, SteREO CL 1500 ECO (Carl Zeiss, Oberkochen, Germany), and the camera system, AxioCam Icc 1 (Carl Zeiss, Oberkochen, Germany), the textures of the fibers were analyzed. The images were processed using the software, AxioVS40 V 4.8.2.0 (Carl Zeiss, Oberkochen, Germany).

### 2.5. Tensile Properties Test

The yarns were tested for tensile strength and elongation using the Texture Analyser TA plus and the associated software, Nextgen Plus, version 3.0 (Ametek, Berwyn, PA, USA). For this purpose, fiber or yarn material was inserted between two clamping blocks spaced 50 mm apart, and the force resulting in a constant pull-out speed of 100 mm/min was measured.

## 3. Results and Discussion

The development of a new individual, temperature-regulated and equipment-specific die for the extrusion of multifilmantous extrudates was made possible using FreeCAD software. To allow the easy cleaning of all the components, a three-part split design was chosen, which can be divided into a two-part 90° deflection and a multifil spinneret (Figure 3). The short-term delivery of these individually manufactured device components was made possible by the cost-effective, additive manufacturing technology within a few days. If a traditional toolmaker who could provide the appropriate certifications for the material and the manufacturing process were able to deliver, the delivery time in 2020 would be 8–10 weeks, compared to a few days for the 3D printing process; in addition, the costs were also about 2.5 to 3 times higher with traditional manufacturing.

For the processing of the extrudates, the table-top conveyor system from Tree-Tec was supplemented by a stainless-steel roller lying on the siliconized draw-off belt. This allowed the extrudate to be stretched while still warm, then passed between the stainless-steel roller and the conveyor belt, before proceeding towards the actual spinning equipment (Figure 4). Due to the extremely strong electrostatic behavior of the individual fine fibers, the direct transfer of the thin fiber materials to the spinning equipment proved to be an extremely decisive step in the development of the manufacturing process in order to exclude the use of the otherwise common spinning oils. The production and handling of the fibers without direct spinning following the extrusion process and without the use of further excipients was not possible in the preliminary trials. For the production of yarns, a motor frequency between 4000 and 5000 Hz proved to be advantageous in the course of development. For the manufacturing of pure fiber materials, the motor frequency needed to be reduced further. Furthermore, the spinning power could be controlled by a commercially available spring brake and, thus, also influence the quality of the fiber and yarn material. To produce yarn-like materials, a brake load of 0.1–0.5 N was found to be preferable. To obtain more fiber-like materials, the braking power was increased further. The power supply for the system was provided by a commercially available laboratory power supply unit.

Various extrusion trials were carried out to gather experience with the handling of the equipment, to characterize it, to find the best parameters using polyvinyl alcohol as an example, and to investigate the properties of the resulting yarns. Due to the multifaceted material properties of the polyvinyl alcohol during the extrusion, various batches of yarns were produced from PVA Parteck MXP, which was specially developed for the hot melt extrusion.

The extrusion temperature of the material was 190 °C, and, therefore, within the range specified by the manufacturer [25]. Lower extrusion temperatures can also be found in the literature [26]. However, extrusion at lower temperatures was not possible due to the temperature-dependent viscosity reduction of PVA [25]. The stretching necessary in this process to particularly low target diameters of the extruded individual fibers required a particularly low viscosity of the extrudate when exiting the die. In addition, the plasticizer glycerol was used in four different concentrations (Setup A–D), and its influence on the extrusion process and the fiber properties was investigated (Table 1).

The extrusion of the fibers was possible at 190 °C and 80 RPM for all the polymer/plasticizer blends listed in Table 1. With increasing plasticizer content, a decrease in torque of about 5 Nm was observed at a constant RPM, between the pure polymer and the blend with 10% plasticizer. In all cases, the fibers were processed with the conveyor system at 4.5 mm/s, and, subsequently, all the fibers were immediately spun at a speed of 4000 Hz, a braking rate of 0.1–0.15 N, and a motor power of 13 W. To investigate the stability of the fiber and yarn materials, the extrudates were stored sealed in multilayer aluminum bags after production until further investigation. To investigate the fiber and yarn materials, a sufficient amount of material was incubated in Petri dishes at 40 °C and 75% r.h., and the stress at break of the materials at different time points after storage was analyzed (Figure 5). In each case, 10 samples were examined at time t = 0 h, 24 h, 48 h, and 168 h.

In the initial investigation of the stress at break, a high force of more than 4 N could be determined for the extrudates without plasticizer, and even 6.5 N for the extrudate with 1% plasticizer. By contrast, no elongation of the yarns was observed, especially in comparison to the yarns with 4.5–10% plasticizer. Here, a lower initial stress at break, with a high extensibility of slightly more than 300% compared to the initial length, was observed. During the storage of the unpackaged extrudates at 40 °C and 75% r.h., it was observed that the extensibility of the yarns with low plasticizer content increased strongly, but the stress at break decreased heavily in the same course. The stress at break decreased within the first 24 h, from 3.3 N to 3.2 N. The further measured values of the samples after 48 h and 168 h slightly increased to 4.2 N. The stress at break and the extensibility of the samples with higher plasticizer content changed only slightly over the storage time. However, some variations within the measurement time point were determined. In particular, the extensibility of the yarns varied to an increased extent in some cases, as shown by the standard deviations.

The examination of the extrudates under the microscope (Figure 6A) showed the occurrence of small air-like inclusions and irregularities at various points across the batches. These inclusions may have resulted from the residual moisture in the starting material or the decomposition of the components in the extruder due to thermal stress. The thermal decomposition of PVA and glycerol is not expected at the temperatures used [27,28]. However, both PVA and glycerol have hygroscopic properties. Therefore, the powder blends were analyzed for residual moisture [25]. The manufacturer, Merck Darmstadt, specifies that the residual moisture of the pure polymer should be less than 0.1% when dried for 3 h at 105 °C. The drying process was carried out at this temperature. To evaluate the drying process, 500 g of PVA Parteck MXP were dried in a hot-air oven, type T 20 (Kendro Laboratory Products, Langenselbold, Germany) for three hours at 105 °C. The samples were obtained at the beginning of drying and every 30 min thereafter. The residual moisture was immediately determined using a residual moisture analyzer at 105 °C (Figure 5B). After three hours, the remaining amount of PVA was taken out of the drying oven and stored at room temperature (22 °C and 52% r.h.), and the residual moisture was continuously monitored using the same method. It was observed that the residual moisture in the polymer decreased continuously over the three hours of drying, but at no time was the residual moisture of 0.1% reached, according to the manufacturer’s specifications. The lowest value, 0.69%, was measured after 150 min of drying (Figure 6B, t = −30 min). During storage at room temperature, the residual moisture of the polymer increased again rapidly. After 60 min, it was already 2.93%, and 170 min after the end of drying, a residual moisture of 5.40% was measured. For pure PVA Parteck MXP, the manufacturer indicates a change in mass at a room humidity above 50% r.h. at 25 °C of 5%, based on the dried mass. At this point, the residual heat of the dried polymer must be taken into account.

The rapid water absorption of the pure polymer could potentially have been enhanced by other hygroscopic materials, such as the plasticizer glycerol used. Due to the high extrusion temperature of the PVA at 190 °C, the evaporation of the absorbed water occurred during compounding. At the spinneret, the expansion of the water vapor could then occur, resulting in the formation of air bubbles in the extrudate. The bubbles were already visually detectable during production and led to the destabilization of the extruded fibers, so that problems with stretching could occur. Such pronounced challenges with moisture are not expected in pharmaceutical melt extrusion, for example, when producing filaments several millimeters in thickness. In particular, the tearing off of the continuous extrudate led to defects in the fiber material and, in addition, the process had to be restarted. For this reason, neither continuous, climate-independent production, nor the production of high-quality particularly thin fiber materials over several hours with potentially hygroscopic materials were possible without individual changes to the manufacturing process.

To enable additional production with hygroscopic materials, the extrusion line was encapsulated. The strong heat generation of the melt extrusion line posed a major challenge for its complete encapsulation, which could have prevented water absorption by the polymer. For this reason, the individual encapsulation of all the potentially critical areas of the production line needed to be ensured. For this purpose, the reservoir of the flat-bottomed feeder was enclosed by means of a 3D printed attachment, which also allowed the introduction of desiccant (Figure 7A). In addition, the continuous monitoring of the room climate inside the flat-bottomed feeder was made possible by means of a commercially available thermohygrometer (Figure 7B). Furthermore, the transition from the flat-bottomed feeder to the entrance of the melt extrusion line was closed with a newly developed attachment individually adapted to the equipment, which also allowed the inclusion of desiccant and the monitoring of the room climate by means of a thermohygrometer (Figure 7A). The use of a transparent top material also made it possible to monitor the filling area to observe any potential material buildup or bridging at an early stage (Figure 7B).

The encapsulation of the system allowed bubble-free extrusion for several hours. In addition, after drying in the drying oven, the polymer was stored in an air-tight environment with the addition of desiccant. Due to the low feeding rate in the described manufacturing process, the flat-bottomed metering unit could remain closed for several hours, and there was no need to open the reservoir to refill the polymer. During several extrusion runs, no bubble-like structures were visually recognizable in the extrudate, and no break-off of the extrudate due to instabilities caused by the air bubbles was observed. The humidity under the encapsulation was consistently below the minimum measurable humidity value of the measuring instruments, of 20% r.h., which is why it was also possible to dispense with changing the desiccant bags during the production runs. This simple and cost-effective form of encapsulation could also prevent potential dust contamination, which could be highly relevant, especially when processing powder mixtures containing active ingredients. An alternative to encapsulation would be the use of a melt extrusion system with automatic degassing of the melted polymer, which is particularly available in larger devices [29].

Light microscopic examinations of the fiber material also showed no extrusion-related air pockets or other inhomogeneities (Figure 8). A uniformly thin and twisted yarn was visible, whose individual fibers were compactly twisted. The light yellowish color of the fiber material corresponded to the usual light coloration of polyvinyl alcohol during thermal processing [25].

In addition to the encapsulation of the production line, the packaging material for the storage of the yarns was also adapted. In order to ensure maximum storage stability, and, in particular, to facilitate potential further processing, the hygroscopic fiber materials were immediately packed and tightly sealed in polyethylene bags with the addition of desiccant. The stress at break of the fiber materials produced and packaged in this way showed a pronounced stress at break, even after more than 12 months of storage in a room climate (Figure 9). The average stress at break of the yarn material determined by the Texture Analyser for a sample of n = 10 was 12.14 N and, thus, significantly higher than the initial values of the short-term stability test. Both the further modification of the manufacturing technique and the packaging seemed to have a positive influence on the mechanical properties of the manufactured material. The low water content, which led to high stress at break in the extrudate, also implied a very low extensibility. A complete drop in force when examining the stress at break of 5 cm of the clamped yarn material in each case could be detected after only 1.1–1.4 mm of distension of the material. The uniform profile of the force path diagrams in Figure 9 indicates a very homogeneous and consistently stable fiber material, which could be produced by means of the novel manufacturing apparatus for pharmaceutically used fiber and thread materials without the use of spinning oil.

## 4. Conclusions

The specific combination of processes already established in pharmaceutical production with the development of novel components enabled the production of pharmaceutically useful fiber and yarn components without the use of other excipients, such as spinning oils. In particular, rapid prototyping enabled the swift and cost-effective production of the required assemblies, which had to be individually manufactured. For the processing of the hygroscopic materials, it was possible and necessary to modify a melt extrusion line used for pharmaceutical purposes. For this purpose, a new type of spinneret in splice design, consisting of a deflection and nozzle plate, was constructed and manufactured using a 3D printing process. This made it possible to change from extrusion in a direction parallel to the ground, which is otherwise common in the pharmaceutical industry, to an extrusion process directed toward the ground. Furthermore, the entire manufacturing line was completely moisture-encapsulated at essential points, preventing water absorption by the extruded material. The significant reduction in air-bubble inhomogeneities in the extrudate that resulted from this led directly to an increase in the resilience of the fiber components and enabled further direct processing. The direct processing of these fiber materials was made possible by the immediate transfer and spinning of the material by the independently developed equipment, Spinning-Rosi. The design of the spinning system and the possibility of the further direct processing of the fiber materials made it possible to dispense with the otherwise customary use of so-called spinning oils, which are used to prevent the development of the electrostatic properties of fiber materials. The yarns spun with the Spinning-Rosi showed no electrostatic behavior due to the twisting, and could be further processed without the aid of additional equipment or excipients. In particular, the choice of packing material can be an important part of the production process for hygroscopic fiber components. In this respect, it was shown that the yarn meshes produced by means of the developed equipment were mechanically stable, even after many months of storage. The use of yarns that are not only particularly flexible but also loaded with active ingredients, such as suture materials, is thus conceivable in the future, and easier to implement from a regulatory perspective. Placebo yarns, such as those used in the innovative EsoCap system for the local placement of films, can also be used. However, new and innovative fiber-based dosage forms are also conceivable from a regulatory perspective.

## Figures and Tables

**Figure 1 pharmaceutics-14-01229-f001:**
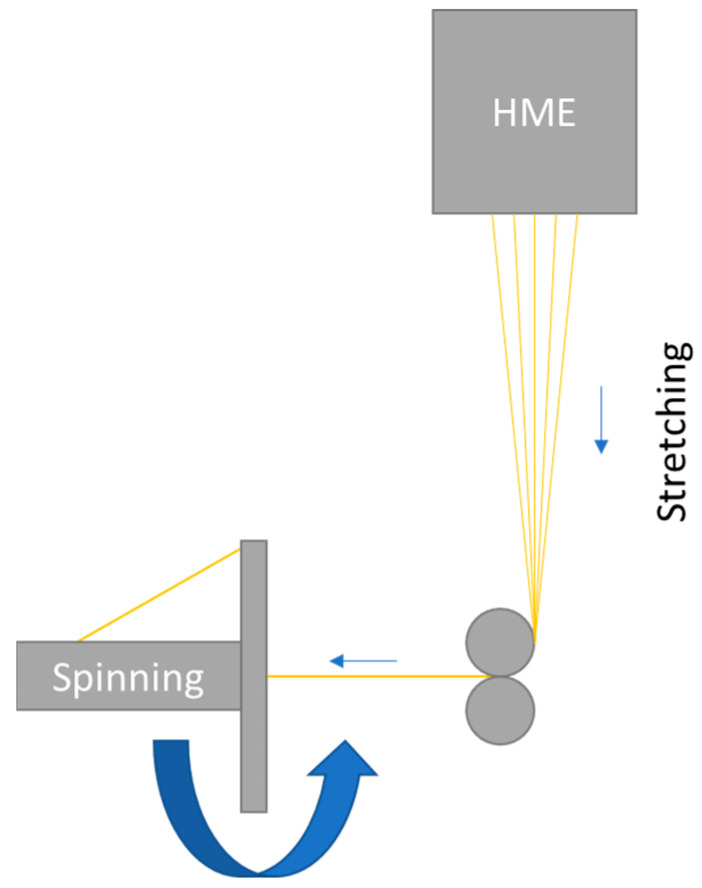
Schematic illustration of the continuous manufacturing design, equipment, and process for pharmaceutically used fibers and yarns: hot melt extrusion (HME) of the polymer used, stretching of the soft polymer into very thin individual fibers, and spinning of the individual fibers into a compact yarn.

**Figure 2 pharmaceutics-14-01229-f002:**
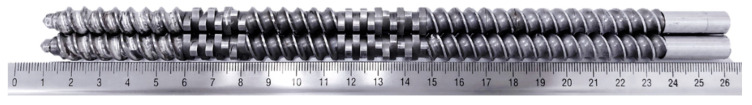
Geometry of the 10-millimeter screw set used in the melt extrusion line.

**Figure 3 pharmaceutics-14-01229-f003:**
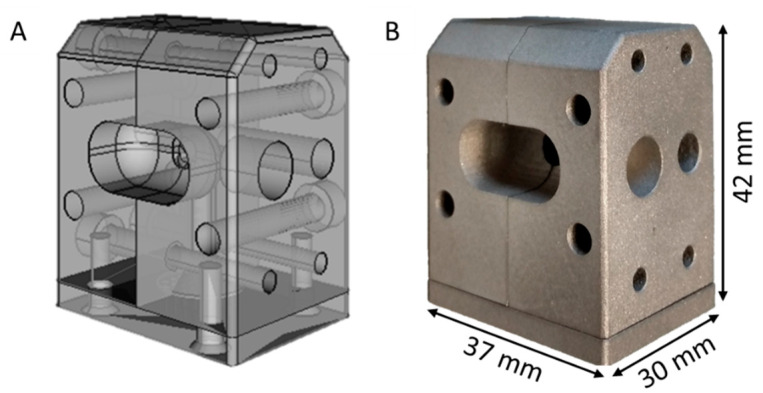
Individually designed spinneret (**A**) manufactured by metal 3D powder bed fusion printing (PBF) (**B**), consisting of the 90° deflection in split design with multifilament die plate for the melt extrusion of pharmaceutical fiber and yarn-like materials.

**Figure 4 pharmaceutics-14-01229-f004:**
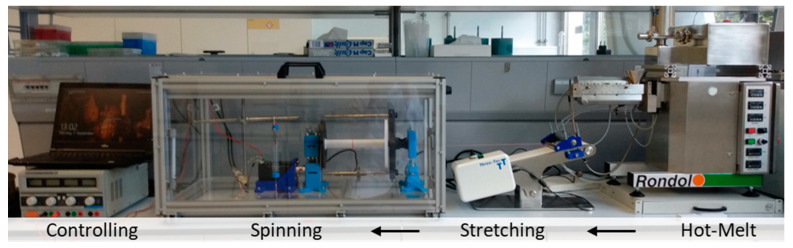
General configuration of the manufacturing setup for the production of pharmaceutical fibers and yarns. From right to left: Hot-melt-extrusion unit with feeding unit, modified table-top conveyor belt, spinning apparatus, power supply, and computer control of the line.

**Figure 5 pharmaceutics-14-01229-f005:**
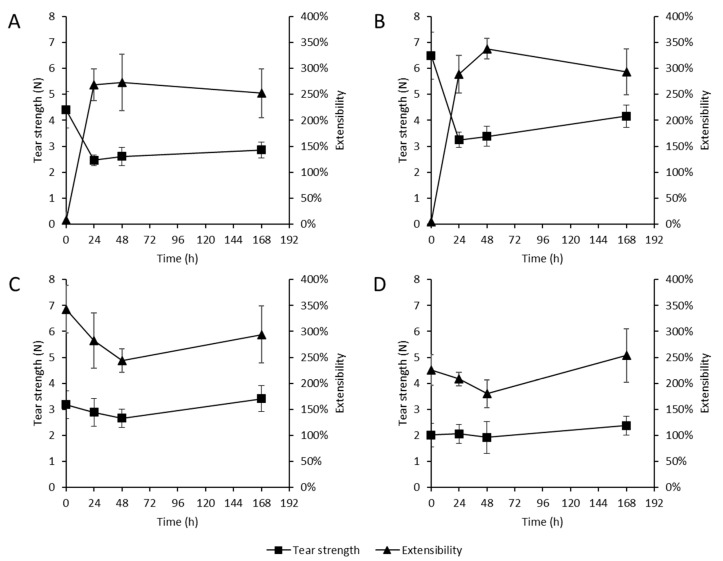
Stress at break (N) and extensibility of multifilament yarns over the time of storage at 40 °C and 75% r.h. (n = 10 +/− SD). (**A**) 100.0% polyvinyl alcohol (PVA); (**B**) 99.0% PVA, 1.0% glycerol; (**C**) 95.5% PVA, 4.5% glycerol; (**D**) 90.0% PVA, 10.0% glycerol.

**Figure 6 pharmaceutics-14-01229-f006:**
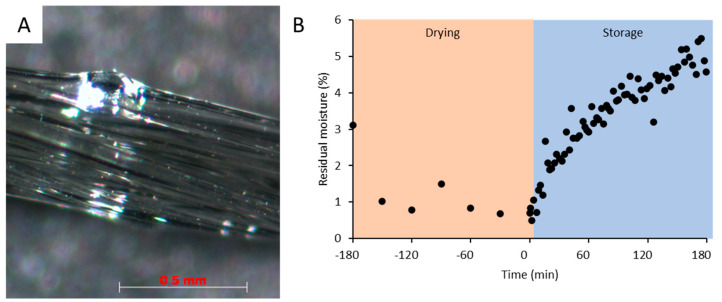
(**A**) Inhomogeneity in the extrudate due to residual water in the polymer during extrusion. (**B**) Residual moisture determination of pure PVA Parteck MXP at 105 °C after 3 h of drying followed by 3 h of storage at room temperature at 52% r.h.

**Figure 7 pharmaceutics-14-01229-f007:**
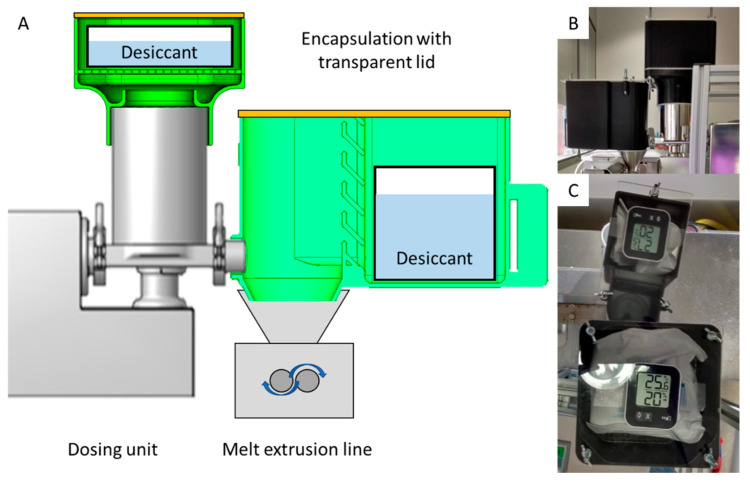
Encapsulation of essential components of a melt extrusion line to prevent water absorption by polymers. (**A**) Schematic illustration of point moisture encapsulation. (**B**) Encapsulation of the flat-bottomed feeder and filling area of the melt extrusion line. (**C**) Encapsulation with introduced desiccant, ambient air monitoring by means of a thermohygrometer, and the possibility of optical monitoring of the filling area of the melt extrusion line.

**Figure 8 pharmaceutics-14-01229-f008:**
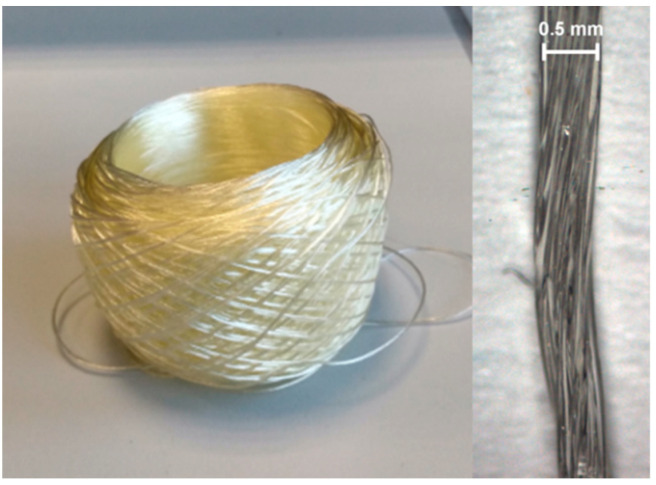
Images of a yarn made from polyvinyl alcohol and glycerol using the Spinning Rosi (from right to left), a spooled roll of the yarn material, and a light microscopy image of the degraded fiber materials.

**Figure 9 pharmaceutics-14-01229-f009:**
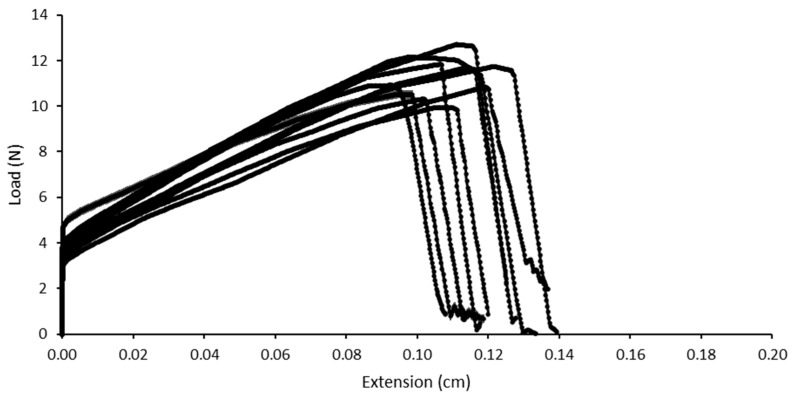
Force-path diagrams of 10 individual samples of a PVA yarn produced using a fully moisture encapsulated melt extrusion system followed by direct spinning. The yarn material was stored for 12 months at room temperature on desiccant packed in polyethylene bags.

**Table 1 pharmaceutics-14-01229-t001:** Composition and feeding rate of the feeder in the production of different batches of fiber and yarn material from polyvinyl alcohol and glycerol.

Materials	Feeding	Setup
PVA Parteck MXP 100.0%	1.5%	A
PVA Parteck MXP 99.0% + Glycerol 1.0%	1.5%	B
PVA Parteck MXP 95.5% + Glycerol 4.5%	1.5%	C
PVA Parteck MXP 90.0% + Glycerol 10.0%	2.0%	D

## Data Availability

The data that support the findings of this study may be available on request from the corresponding author W.W., depending on requested information.

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
