# Peer review of "Development of a Hot-Melt-Extrusion-Based Spinning Process to Produce Pharmaceutical Fibers and Yarns"

_pharmaceutics, 2022, doi:10.3390/pharmaceutics14061229_

Round 1

Reviewer 1 Report

The manuscript describes the development and validation of a new fibre spinning production line that does not require the use of oils during the spinning process. The results are interesting and should be published after some minor corrections.

  1. In the introduction, please be aware that 3D printing may or may not be the most cost-effective way to manufacture a prototype for your fibre spinning device, depending on the complexity and the material requirements. 
  2. Did you compare machining to 3D printing in terms of costs when deciding to go for 3D printing? If this cost analysis was done, please mention it in the manuscript.
  3. In Figure 1, please define what HME stands for either on the drawing or at least in the caption. HME is not defined anywhere in the manuscript.
  4. In materials and methods, please explain how was the "polymer rubbed with a plasticizer." What device was used for this? 
  5. The word "quality" in line 108 is too ambiguous. Please be more specific. What level of moisture and temperature are required for ensuring acceptable extrusion results?
  6. In lines 108-109, please define "the required components." Required components for what exactly? 
  7. In line 117, please define what kind of 3D printing was used for fabricating the spinneret. (PBF, MEX, BJT, DED, MJT, SHL, VPP?) I suggest using the standard names as defined by: https://www.iso.org/obp/ui/#iso:std:iso-astm:52900:dis:ed-2:v1:en
  8. In line 126, please replace "anin-house" with "an in-house."
  9. The tensile test performed does not measure the tear strength (e.g. https://www.iso.org/obp/ui/#iso:std:iso:34:-1:ed-4:v1:en). Based on the description provided, you measure the tensile properties of the yarn, and the force at break is being reported. Tear strength is related to fracture propagation measurements, which were not done. Please correct the title of section 2.5.
  10. In line 157, please specify what type of 3D printing was used to make the spinneret (e.g., powder bed fusion (PBF)). 
  11. In Figure 4, please label the components in the figure. Labelling will make it more evident than the description provided in the caption.
  12. In line 196, please measure and provide the speed in mm/s of the level 3 speed of the used conveyor belt.
  13. In Figure 5, please report the stress at break in Pa, not the force in N. 
  14. In line 226, the word "described" does not fit the context, "expected" could be better suited. Please also cite a paper where the degradation temperature of PVA and glycol are provided.
  15. Please explain what the negative time means in Figure 5B. I suggest labelling the graph drying time and storage time rather than negative times. It is not very clear.
  16. Please label the components in Figure 7A and provide a diagram where the setup can be seen. Where is the polymer, and where is the desiccant? Are they in contact?
  17. Line 308 and 309 and throughout the manuscript, please replace "tear strength" with "stress at break."
  18. In the conclusion section, it is stated that "The specific combination of the arrangement of techniques already established in the production of pharmaceutical products with the development of novel components using modern additive manufacturing technologies ...". Too much emphasis is placed on the fact that some components of the production line were fabricated by AM technologies. However, the manuscript does not show that AM is necessary to achieve the results presented (i.e. the geometry of the spinneret and encapsulation components can only be achieved by AM?). The manuscript also did not show that AM is cheaper than machining. Therefore, this sentence should be rephrased to mention the real advantage of using AM during the development of the fibre spinning device. 

Author Response

Dear Reviewer, first of all, we would like to thank you for the valuable comments. We responded in a detailed manner to all comments. Where appropriate, we changed the manuscript accordingly. Changes in the manuscript are highlighted according to the journal's specifications using the "Track Changes" feature and are additionally shown in the following document.

Reviewer 2 Report

The authors have prepared HME based fibers of PVA by varying the concentration of glycerol. I could not find any novelty in work. Moreover, the research work is carried out for adressing certain aims and objectives. The ultimate application of the work must be defined and experimetally proved. Just preparing fibers doenot satisfy the novelty of the work

Author Response

(The authors gave the same response as above.)

Reviewer 3 Report

Dear authors,

thanks a lot for submitting this interesting paper. While I found the paper overall quite informative and well structured, I think it can be improved in some parts.

1) Abstract: I propose to re-write the abstract to add more clarity and to make the paper more attractive for the reader. Keep the background short, e.g. there are technical processes for producing fibres as well as pharmaceutical applications, but certain hurdles need to be overcome (oil) in order to use these processes in a pharmaceutical manufacturing. Than focus more on what you have done in the work.

2) Line 98 to 118: This is actually one of the most important parts of the paper, where you describe the set-up of the equipment and the experiments. You start the paragraph good by dividing the process in three areas, but afterwards it is difficult to understand for the reader how the set-up is. It seems to be mixed with information on how these additional parts were produced. It would be good if you can decouple it, e.g. first describe the set-up and than the production of the parts.

3) Conclusion: I miss an outlook here. What needs to be done in addition to make the process applicable to Pharma manufacturing? Are additional trials planned to include a drug substance? ...

Furthermore, the following lines might be rephrased and/or typos removed:

Line 41: "particular suitable" sounds actually very positive, which does not fit to the first part of the sentence where you describe the throughput rather as limitation. In addition you use the same phrase in the next sentence again.

Line 126: Missing space between "an in-house".

Line 168: Manufacturing instead of Manufaction?

Line 179: What is meant by system (the formulation or the experimental set-up)?

Line 234 & 240: Wrong reference. Instead of Figure 5B it should be 6B.

Author Response

(The authors gave the same response as above.)

Reviewer 4 Report

Pharmaceutics

Development of a hot melt extrusion based spinning process to 2 produce pharmaceutical fibers and yarns

Pharmaceutics-1696571

Comments and Questions

Several new approaches are made to modify a melt extrusion line used for pharmaceutical purposes for the processing of hygroscopic materials such as polyvinyl alcohol and the plasticizer glycerol used:

  • A new type of spinneret in splice design consisting of a deflection and nozzle plate was developed and manufactured using the 3D printing process. The extrusion direction parallel to the ground, which is common in the pharmaceutical industry, is changed to an extrusion process directed toward the ground,
  • The entire manufacturing line was completely moisture encapsulated at essential points, preventing water absorption by the extruded material,
  • The direct processing of these fiber materials was immediately transferred and spin by the independently developed equipment called Spinning- Rosi without the need of the customary use of spinning oils, which are intended to prevent electrostatic properties of the fiber material. The yarns spun with the Spinning-Rosi showed no electrostatic behavior due to the twisting and could be further processed without the aid of additional equipment or auxiliary materials,
  • In particular, the choice of packing material can be an important part of the production process for hygroscopic fiber components. In this respect, it could be shown that the yarn materials produced by means of the developed equipment represent mechanically stable meshes even after many months of storage.

This interesting study can be published after answering the following questions:

  • What are the “multifaceted material properties” of PVA exactly mean and include in Line 180? Are those the reasons why other biocompatible and self-degradable polymers such as chitosan and PLA are not studied here?
  • The dimensions and scale of the spinneret in Figure 3 were not given.
  • Should the drying behavior like the one in Figure 6b for pure PVA Parteck MXP be studied again for the yarn materials indicated in Figure 8 made from polyvinyl alcohol and glycerol using the Spinning Rosi?
  • Why fiber-based dosage form was not demonstrated here with the addition of a model active pharmaceutical ingredient?
  • In Line 234, “(Figure 5B)” should be changed to “(Figure 6B)”.
  • What are the water adsorption mechanisms? Can the kinetics be revealed with FTIR or TGA?  Could XPS be used to show how PVA, glycerol and water interact with one another?  Could molecular drawings be added to explain the molecular structure-extensibility/strength relationships in Figures 5 and 9 for the roles of PVA, glycerol and water?

Author Response

(The authors gave the same response as above.)

Round 2

Reviewer 2 Report

The work seems to be of industrial consultacy work for developing pharmaceutical grade fibers. However, number of reported literature is available where fibers have been prepared using the same composition.

Moreover, due to the lack of application based approach, the work may not be useful for the readers of the journal.

Author Response

Thank you very much for your feedback. We have revised the manuscript again in various places and also highlighted various places in the document which, from our point of view, explain the novelties of the bearings described in the manuscript. These include in particular the combination of different systems described so far and the modification of these assemblies using rapid prototyping to create a continuous spinning process. It is this continuous spinning process that enables us to handle the particularly thin individual fibers of the extrudate in the first place, as these can only be processed with spinning oils in the classic multi-step process due to their static repulsion.

The entire concept was developed entirely at the university. All components have been self- constructed with a freely available CAD software and show in combination with the used 3D printing its potential in the development of small batch sizes.

The polyvinyl alcohol filaments produced have, as described, a direct use as a rapidly soluble filament in the EsoCap concept of a novel dosage form developed in the working group for local application of films in the esophagus. In the future, we are pleased to have the technical capability to produce pharmaceutical fibers and to develop further novel drug-loaded dosage forms from these fibers. Due to the newly constructed punctual encapsulation, it is now possible for us to produce particularly hygroscopic fibers.

In particular, the aspect that our project partner had contacted a number of companies (suture and melt extrusion) and institutes (various German competence centers for fiber and textile technology) in advance with the request to produce a multifilament yarn from polyvinyl alcohol without the use of a spinning oil, unfortunately without success, demonstrates from our perspective the novelty of the system we have shown.

With our comments and explanations, we hope to be able to present the novelties and challenges in the context of our development work.

Reviewer 4 Report

Accept in present form.

Author Response

Thank you for the positive feedback and your constructive comments during the review process.